# Subcritical Methanol Extraction of the Stone of Japanese Apricot *Prunus mume* Sieb. et Zucc.

**DOI:** 10.3390/biom10071047

**Published:** 2020-07-15

**Authors:** Tomoaki Kawabata, Yoshie Tanaka, Asako Horinishi, Megumi Mori, Asao Hosoda, Nami Yamamoto, Takahiko Mitani

**Affiliations:** 1Faculty of Biology-oriented Science and Technology, Kindai University, Kinokawa, Wakayama 649-6493, Japan; cow13utter@gmail.com (T.K.); horinishi@waka.kindai.ac.jp (A.H.); 2School of Medicine, Department of Chemistry, Wakayama Medical University, Wakayama 641-0011, Japan; tanaka@wakayama-med.ac.jp; 3Industrial Technology Center of Wakayama Prefecture, Wakayama 649-6261, Japan; mmori@wakayama-kg.jp (M.M.); hosoda@wakayama-kg.jp (A.H.); 4Faculty of Education, Wakayama University, Wakayama 640-8510, Japan; namiyama@wakayama-u.ac.jp; 5Center of Regional Revitalization, Research Center for Food and Agriculture, Wakayama University, Wakayama 640-8510, Japan

**Keywords:** *Prunus mume*, Japanese apricot, mume stones, subcritical methanol extraction, coniferyl alcohol, sinapyl alcohol

## Abstract

The pits of Japanese apricot, *Prunus mume* Sieb. et Zucc., which are composed of stones, husks, kernels, and seeds, are unused by-products of the processing industry in Japan. The processing of Japanese apricot fruits generates huge amounts of waste pits, which are disposed of in landfills or, to a lesser extent, burned to form charcoal. Mume stones mainly consist of cellulose, hemicellulose, and lignin. Herein, we attempted to solubilize the wood-like carapace (stone) encasing the pit by subcritical fluid extraction with the aim of extracting useful chemicals. The characteristics of the main phenolic constituents were elucidated by liquid chromatography-mass spectrometry (LC-MS) and nuclear magnetic resonance (NMR) analyses. The degrees of solubility for various treatments (190 °C; 3 h) were determined as follows: subcritical water (54.9%), subcritical 50% methanol (65.5%), subcritical 90% methanol (37.6%), subcritical methanol (23.6%), and subcritical isopropyl alcohol (14.4%). Syringaldehyde, sinapyl alcohol, coniferyl alcohol methyl ether, sinapyl alcohol methyl ether, 5-(hydroxymethyl)-2-furfural, and furfural were present in the subcritical 90% methanol extract. Coniferyl and sinapyl alcohols (monolignols) are source materials for the biosynthesis of lignin, and syringaldehyde occur in trace amounts in wood. Our current findings provide a solubilization method that allows the main phenolic constituents of the pits to be extracted under mild conditions. This technique for obtaining subcritical extracts shows great potential for further applications.

## 1. Introduction

Fruits and their industries generate large quantities of waste and by-products, and the management of waste and by-products is one of the major environmental issues [1,2,3,4]. On the other hand, recent research has showed that it is rich in potentially valuable components such as polyphenol, pigments, and vitamins in the waste and by-products. Applying various extraction and purification techniques, it has been possible to recover the important components from them [1,2,3,4].

The Japanese apricot, *Prunus mume* Sieb. et Zucc., belongs to the Rosaceae family and is one of the most popular fruit trees in Japan. Unlike many other fruits, mume fruit is extremely sour owing to the accumulation of citric acid, with the acidity of mature fruit reaching up to 6–7% (flesh weight). Therefore, most harvested mume fruit is processed into umeboshi (pickled mume fruit), which is very popular in Japan. During the umeboshi production process, any misshapen umeboshi are processed into mume paste, affording the pit as a by-product. Thus, this industry has led to the production of vast amounts of pit waste.

The pit of *Prunus* fruits is encased in an extremely hard wood-like carapace called the stone. It was clarified in the early 1960s that peach stones contain lignin [5,6]. As such stones consist of woody material (cellulose, hemicellulose, and lignin), practical solubilization or degradation methods are key for their chemical analysis and further utilization. Thus, to promote the effective use of pits as plant biomass, many treatments have been tested to date. However, a satisfactory and useful method still has not been found.

There are numerous methods used to extract bioactive compounds from natural sources. As green chemistry for extraction purposes has progressed, various extraction methods have become available recently: ultrasound-assisted extraction, microwave-assisted extraction, supercritical fluid extraction, mechanical pressing, and instant controlled pressure-drop (DIC) extraction [7]. Mume stones were auto-hydrolyzed by microwave heating to extract polysaccharides and phenolic compounds [8]. By heating at 200–230 °C, 48.0–60.8% of the polysaccharides and 84.1–97.9% of the phenolic compounds were extracted in water; however, this treatment caused the extensive degradation of hemicelluloses and lignin. The extracted liquors exhibited antioxidant activity against hydroxyl and DPPH radicals originating from the phenolic compounds. Under these conditions, considerable amounts of carbohydrate and phenolic compounds were extracted from the mume stones at 230 °C. However, brownish discoloration attributed to extensive degradation of the phenolic compounds was observed at this temperature. Therefore, it is likely that microwave heating is associated with the mass degradation of phenolic constituents in stone samples. The stability of 22 compounds from different families of phenolic compounds was investigated under microwave-assisted extraction [9]. It was found that all the compounds are stable at temperatures <100 °C, and that compounds with a large number of hydroxy-type substituents degrade more easily at 125 °C; treatment at 230 °C resulted in significant degradation of the native and active compounds. Hence, to isolate the native compounds from fruit stones, a milder extraction treatment must be adopted.

The aim of this study was to develop a milder and more effective method to dissolve mume stones. Therefore, we investigated the use of subcritical fluid extraction and attempted to clarify the characteristics of the main phenolic constituents in mume stones.

## 2. Materials and Methods

### 2.1. Reagents

Gallic acid monohydrate, caffeic acid, *p*-coumaric acid, ferulic acid, and sinapyl alcohol were purchased from Sigma Aldrich (St. Louis, MO, USA). Coniferyl alcohol was acquired from Extrasynthese (Genay, France). Furfural and 5-(hydroxymethyl)-2-furfural were obtained from Nacalai Chemicals (Kyoto, Japan). HPLC-grade methanol and trifluoroacetic acid (TFA) were purchased from Wako Chemicals (Tokyo, Japan). All other chemicals used were of analytical grade.

### 2.2. Preparation of Mume Stones

Fruit samples of *P. mume* “Nanko” were randomly collected from one specific tree grown at the experimental orchard of the Laboratory of the Japanese Apricot, Fruit Tree Experiment Station of the Wakayama prefecture government. Samples were washed with tap water and dried with tissue paper. Subsequently, the flesh and seeds of the fruit were separated. The seeds were further divided into the mume stone, internal seed coat, and albumen; the remaining flesh was removed from the mume stones by brushing. The stones were dried in a vacuum desiccator for 2 days and were subsequently milled using a Wonder Blender WB-1 (Osaka Chemical Co. Ltd., Osaka, Japan). Next, the milled mume stones were passed through a sieve (mesh size no. 100); the particle size was less than 2 mm. The sifted powder was washed once with acetone and dried under air at room temperature. The resultant powder (10 g) was suspended in *n*-hexane (25 mL) and sonicated for 2 min. The suspension was then centrifuged for 5 min at 3000 rpm, and the resultant residue was washed twice with *n*-hexane (25 mL) and dried under air at room temperature. During lipid extraction, the weight decreased by <2%. This residue was labeled as HTRS.

### 2.3. Isolation of Lignin

Lignin, one of the main components in wood, was analyzed using the official method of the Japan Wood Research Society [10]. Briefly, three HTRS samples (1.0 g each) were added to 15 mL of H_2_SO_4_ solution (72%, *w*/*w*) and stirred occasionally over 4 h. Next, 560 mL of water was added to the suspension. The resultant mixture was heated using reflux at 100 °C for 4 h and the remaining suspension was filtered through a glass microfiber filter. The solid residue was washed with hot/cold water, dried at 105 °C, and weighed (acid-insoluble lignin). Additionally, the absorbance, A (205–210 nm), of the filtrate was measured by using the UV–Vis spectrophotometer UV-1800 (Shimadzu Co., Kyoto, Japan) and the amount of acid-soluble lignin (ASL) was determined using equation 1, where the absorption coefficient of lignin (ACL) = 110 L/g.cm.
ASL (%) = 100 × filtrate (L) × (A of filtrate − A of 3% H_2_SO_4_)/110 × sample (g)(1)

### 2.4. Extraction with Methanol, Ethanol, Isopropyl Alcohol, or Acetone

HTRS samples (500 mg) were suspended in 25 mL of methanol, ethanol, isopropyl alcohol, or acetone, and heated at 60 °C for 3 h. Each suspension was centrifuged for 5 min at 3000 rpm at room temperature, and the residual fraction was washed two times with the corresponding solvent. The resultant residue was dried at 90 °C and then weighed.

### 2.5. Acid Extraction

HTRS samples (500 mg) were treated with 4 M TFA at 90 °C for 1–3 h. After extraction, the supernatant was collected and concentrated by rotary evaporation. The solvent-insoluble residue was washed with water and dried at 90 °C.

### 2.6. Extraction with Subcritical Water, Methanol, or Isopropyl Alcohol

HTRS samples (2.5 g) were charged into a 90 mL stainless steel batch reactor (Taiatsu Techno Corp., Tokyo, Japan) equipped with a supercritical extraction reactor. Next, 50 mL of solvent (distilled water, 50% methanol, 90% methanol, methanol, or isopropyl alcohol; degassed before mixing) was added. The reactor was heated to the desired temperature (190 °C) and subsequently filled with the required solvent via a solution sending pump; the pressure in the reactor was maintained at 10 MPa (i.e., 100 bar) for 3 h. The reactor was then cooled to room temperature and its contents were collected and centrifuged for 5 min at 3000 rpm. The supernatant was concentrated by rotary evaporation and freeze-dried, whereas the residue was dried at 90 °C and then weighed.

### 2.7. HPLC Analysis

A 5 µL aliquot of extract solution was separated using an Agilent HPLC system (Santa Clara, CA, USA) equipped with a 5 µm Hypersil GOLD column (4.6 mm I.D. 150 mm; Thermo Scientific, Waltham, MA, USA). All analyses were monitored by a UV detector at 280 nm. Linear gradient elution was performed at a flow rate of 1.0 mL/min and a temperature of 40 °C using solvents A (0.1% TFA) and B (methanol) in the following sequence: 20% B in A; 20–25% B in A, 3–20 min; 25–45% B in A, 20–40 min; 45–100% B in A, 40–45 min.

### 2.8. Preparative HPLC

An AKTA Explorer 10S system (GE Healthcare Bioscience, Marlborough, MA, USA) was used for preparative HPLC. A portion of the sample was loaded onto a Hydrosphere C18 column (10 × 250 mm, 5 µm; YMC, Kyoto, Japan). Separation was achieved by elution with a series of linear gradients of solvents C (0.1% formic acid in water) and D (methanol) at a constant temperature of 4 °C and a flow rate of 1.5 mL/min, as follows: 15% D in C; 15–35% D in C, 0–20 min; 35–100% D in C, 220–260 min. Aliquots of eluates corresponding to each peak were collected, evaporated to dryness, and re-dissolved in methanol.

### 2.9. LC-MS and NMR Analysis

Liquid chromatography–mass spectrometry (LC-MS) analyses were performed by electrospray ionization (ESI)-MS with an Exactive mass spectrometer (Thermo Scientific) interfaced with an Accela 600 HPLC (Thermo Scientific). A portion of the sample was loaded onto a Hydrosphere C18 column (4.6 mm, 250 mm, 5 µm; YMC, Kyoto, Japan). Separation was achieved by elution with acetonitrile/0.5% acetic acid aqueous solution (1:1, *v*/*v*) at a constant temperature of 30 °C and a flow rate of 1.0 mL/min.

Nuclear magnetic resonance (NMR) spectra were recorded on an ANANCE 400 NMR spectrometer (Bruker Biospin, Billerica, MA, USA). Chemical shifts were reported as δ values (ppm) relative to internal tetramethylsilane (TMS) and the coupling constants (*J*) were given in Hz.

### 2.10. Spectral Data of Isolated Compounds

Compound III (syringaldehyde); IUPAC name: 4-hydroxy-3,5-dimethoxybenzaldehyde; formula: C_9_H_10_O_4_; mass: 182.1733; exact mass: 182.057908808. ^1^H-NMR (400 MHz, acetone-*d*_6_, TMS) δ 9.82 (1H, s), 7.24 (2H, s), 3.92 (6H, s). ESI-FTMS, *m*/*z* 183.0653 [M + H]^+^.

Compound IV (sinapyl alcohol); IUPAC name: 4-[(1*E*)-3-hydroxyprop-1-en-1-yl]-2,6-dimethoxyphenol; formula: C_11_H_14_O_4_; mass: 210.2265; exact mass: 210.089208936. ^1^H-NMR (400 MHz, acetone-*d*_6_, TMS) δ 6.73 (2H, s), 6.49 (2H, dt, *J* = 16 Hz), 6.25 (1H, dt, *J* = 4 Hz, 16 Hz), 4.20 (2H, dd, *J* = 4 Hz), 3.84 (6H, s), 3.82 (6H, t). ESI-FTMS, *m*/*z* 211.0967 [M + H]^+^.

Compound V (coniferyl alcohol methyl ether); IUPAC name: 2-methoxy-4-[(1*E*)-3-methoxyprop-1-en-1-yl]phenol; formula: C_11_H_14_O_3_; mass: 194.2271; exact mass: 194.094294314. ^1^H-NMR (400 MHz, acetone-*d*_6_, TMS) δ 7.66 (1H, brs), 7.09 (1H, d), 6.89 (1H, dd, *J* = 4 Hz, 4 Hz), 6.78 (1H, d, *J* = 8 Hz), 6.52 (1H, dt), 6.16 (1H, dt), 4.01 (2H, dd, *J* = 4 Hz, 4 Hz), 3.87 (3H, s), 3.29 (3H, s). ESI-FTMS, *m*/*z* 193.0860 [M - OCH_3_ + H]^+^.

Compound VI (sinapyl alcohol methyl ether); IUPAC name: 2,6-dimethoxy-4- (3-methoxyprop-1-en-1-yl)phenol; formula: C_12_H_16_O_4_; mass: 224.253; exact mass: 224.104859. ^1^H-NMR (400 MHz, acetone-*d*_6_, TMS) δ 7.30 (1H, brs), 6.76 (2H, s), 6.51 (1H, dt, *J* = 16 Hz), 6.18 (1H, dt, *J* = 4 Hz, 16 Hz), 4.01 (2H, dd, *J* = 4 Hz, 4 Hz), 3.84 (6H, s), 3.30 (3H, s).ESI-FTMS, *m*/*z* 247.0944 [M + Na]^+^.

## 3. Results

### 3.1. Lignin Contents in Mume Stones

The total lignin content was determined to be 46.5 ± 0.3% (acid-insoluble lignin = 46.4 ± 0.2% and acid-soluble lignin = 0.14 ± 0.0%).

### 3.2. Extraction with Methanol, Ethanol, Isopropyl Alcohol, or Acetone

Extraction of the HTRS sample by various solvents (methanol, ethanol, isopropyl alcohol, or acetone) was attempted by heating to reflux for 3 h. The mume stone samples were insoluble or slightly soluble in these solvents under ordinary pressure (Figure 1). HPLC analysis of each soluble fraction (Figure 2) revealed early eluting peaks; however, owing to the low reactivity with the Folin–Ciocalteu reagent [11], these peaks were not attributed to phenolic compounds. Therefore, it was suggested that these extraction procedures were not effective to solubilize the woody parts of the mume stones.

### 3.3. Acid Extraction

Approximately 40–50% of each HTRS sample was solubilized or decomposed on treatment with 4 M TFA (Figure 1). HPLC analysis of the soluble fraction revealed several peaks at retention times (RTs) between 2 and 10 min (Figure 3). The peaks at RTs of 2.8 and 3.6 min were named peaks I and II, respectively. The co-chromatogram demonstrated that the RTs of peaks I and II were identical to those of furfural and 5-(hydroxymethyl)-2-furfural, respectively. Additionally, MS analysis indicated that the *m*/*z* ratio of peak I was 126.11, which is similar to the molecular weight of 5-(hydroxymethyl)-2-furfural. However, no phenolics were observed in the chromatogram of the TFA-soluble fraction. We thought that the treatment with TFA decomposed cellulose and hemicellulose in the stones, but not lignan.

### 3.4. Extraction with Subcritical Water, Methanol, or Isopropyl Alcohol

The HTRS sample was treated with subcritical water, subcritical 50% methanol, subcritical 90% methanol, subcritical methanol, or subcritical isopropyl alcohol for 3 h. As summarized in Figure 1, the effect of each treatment on the degree of solubility was as follows: subcritical water (54.9%), subcritical 50% methanol (65.5%), subcritical 90% methanol (37.6%), subcritical methanol (23.9%), and subcritical isopropyl alcohol (14.4%). The HPLC chromatogram of each solubilized fraction is illustrated in Figure 4. In the early hours of subcritical water extraction, several peaks were observed at RTs of 2–9 min, which were labeled peaks I, II, and III (RT = 8.7 min). Increasing the methanol ratio in the subcritical solvent resulted in a decrease in the heights of peaks I and II, but an increase in that of peak III. Furthermore, peaks IV (RT = 10.0 min), V (RT = 29.4 min), and VI (RT = 31.0 min) appeared and increased over time (Figure 5). Peaks V and VI were not observed in the subcritical isopropyl alcohol extract.

The RTs of peaks I and II were coincident with those of 5-(hydroxymethyl)-2-furfural and furfural, respectively. To isolate the compounds corresponding to peaks III, IV, V, and VI, the subcritical methanol extracts were washed with *n*-hexane and the methanol phase was condensed. Then, the samples were separated by preparative HPLC and analyzed by HPLC (Figure 6), and each compound was identified by NMR analysis. The MS and NMR spectra of each isolated compound were shown in Materials and Methods. Peaks III, IV, V, and VI were attributed to syringaldehyde (4-hydroxy-3,5-dimethoxybenzaldehyde), sinapyl alcohol (4-[(1*E*)-3-hydroxyprop-1-en-1-yl]-2,6-dimethoxyphenol), coniferyl alcohol methyl ether (2-methoxy-4-[(1*E*)-3-methoxyprop-1-en-1-yl] phenol), and sinapyl alcohol methyl ether (2,6-dimethoxy-4-[3-methoxyprop-1-en-1-yl] phenol), respectively (Figure 7). In Figure 4C,E,F and Figure 5, there is an unknown peak (RT = 4.34), which seems like a hydrophilic substance. We analyzed several candidate compounds by HPLC: protocatechuic acid, benzoic acid, caffeic acid, syringic acid, and vanillin. However, there were no corresponding compounds which have the same RT. The identification of this peak is a remaining subject.

### 3.5. Methylation During Subcritical Methanol Extraction

In the subcritical methanol extracts of HTRS, an increase in sinapyl alcohol methyl ether and coniferyl alcohol methyl ether was observed as the reaction time increased (Figure 8). The methylation reaction is thought to proceed over time. To confirm this, we treated authentic sinapyl and coniferyl alcohols with subcritical methanol, and found that both alcohols were successfully methylated.

## 4. Discussion

Fruits after processing generate a huge volume of waste, which are in the form of peel, pulp, seed, and stem, etc. Most fruit wastes have been utilized as a source of livestock feeds or as organic fertilizers. Since non-edible parts of fruit such as peel, skin parts, and seed often contain higher amounts of bioactive compounds than the edible parts, it has attracted attention to obtain the bioactive compounds in the wastes [1,2,3,4]. One of the most common target compounds from fruit wastes is polyphenols. The suitable and frequently used methods for isolation of polyphenols are solvent extraction involving pressure and microwaves.

As the stone of mume is extremely hard, we found that it was impossible to solubilize the stone by conventional organic solvents such as methanol, ethanol, isopropyl alcohol, and acetone.

Supercritical fluid extraction also has been widely studied for the extraction of polyphenols from natural plant materials [12]. The supercritical fluid extraction may be grouped into five methods based on a type of fluid: (a) pure and aqueous ethanol, (b) water, (c) sequential extraction with carbon dioxide, ethanol, and water, (d) methanol, and (e) other solvents (hexane, ethyl acetate, ethyl lactate, isopropyl alcohol). In addition to the type of fluid, pressure and temperature conditions are important. Although there are many studies on the separation of phenolic compounds with supercritical carbon dioxide in the temperature range 40–60 °C and pressure range 100–400 bar, it was thought that the superfluid extraction of mume stones with carbon dioxide was not efficient by reason of the hardness of the stones. Therefore, we selected ethanol, water, methanol, and isopropyl alcohol as a subcritical fluid.

It was shown that extractions with subcritical methanol or aqueous methanol were found to provide more efficient extraction of the phenolic compounds in mume stones, although the extraction was accompanied with a methylation reaction. By contrast, subcritical water exhibited the ability to oxidize various materials via wet oxidation processes, resulting in the rapid oxidation of extracted organic compounds. A large amount of furfural was detected in the extracts afforded from subcritical water extraction. It is known that furfural may be obtained by the acid catalyzed dehydration of pentoses which are present in the hemicellulose of biomass. In these conditions, therefore, subcritical water acts as a strong acid and decomposes the cellulose and lignin present in the mume stone. This also suggests that increasing the methanol ratio in the subcritical aqueous methanol resulted in a decrease in furfural. During subcritical methanol extraction, the amount of methylated sinapyl and coniferyl alcohols increased as the methylation reaction proceeded. Purified sinapyl and coniferyl alcohols were also methylated in the subcritical methanol extraction. It is known that lignin of dicotyledonous angiosperms is consisted of coniferyl alcohol and sinapyl alcohol [13]. Therefore, it seems likely that lignin of mume stone has mainly coniferyl alcohol, and sinapyl alcohol. However, as the maximum durable temperature of our superfluid equipment is 190 °C, we could not test at a higher temperature than 190 °C. It is necessary to attempt to test at higher temperature by using other equipment. On the other hand, to avoid degradation of phenol compounds, subcritical methanol extraction at lower temperature for a short period of time during the extraction process was tried preliminarily. It was clarified that the solubility of stones was significantly decreased.

Subcritical water was used to hydrolyze rice bran and release phenolic compounds; however, the high temperatures used in this extraction process also caused the decomposition of phenolic acids [14]. Thermal analysis of the phenolic acids in the solvents demonstrated that *p*-coumaric, caffeic, and ferulic acids started to decompose at ~170 °C, whereas gallic acid did not start to decompose until a temperature of ~200 °C. The natural products are affected by thermal conditions and the high possibility of degradation occurring during all kinds of processing steps.

Recently, mume stones were auto-hydrolyzed by microwave heating to extract polysaccharide and phenolic compounds [8]. Although considerable amounts of carbohydrate and phenolic compounds were extracted from the stones at 230 °C, extensive degradation of the phenolic compounds also occurred. These identified compounds were benzoic acid, vanillin, syringaldehyde, vanillic acid, syringic acid, and sinapyl aldehyde. It was suggested that oxidative decomposition of coniferyl alcohol and sinapyl alcohol occurred during microwave heating.

In this paper, we found that syringaldehyde, sinapyl alcohol, coniferyl alcohol methyl ether, sinapyl alcohol methyl ether, 5-(hydroxymethyl)-2-furfural, and furfural were present in the subcritical 90% methanol extract of mume stones. It has been reported that syringaldehyde moderately inhibited cyclooxygenase-2 activity [15], although it displayed no antioxidant activity. Syringaldehyde had an antihyperglycemic effect in streptozotocin-induced diabetic rats [16]. Furthermore, coniferyl alcohol inhibits the growth of *Verticillium longisporum*, a soil-borne vascular pathogenic fungus, in vitro, but no such inhibition occurs with sinapyl alcohol [17]. Although many studies on the roles of sinapyl and coniferyl alcohols in the biosynthesis of lignin are available, pharmacological research on these alcohols is rare. Therefore, using our proposed extraction technique, it may be possible to develop subcritical solvent extracts from mume stones for application in pharmacological research.

Cyanogenic glycosides such as amygdalin and prunasin are present in Japanese apricot. In Japan, most mume fruit is processed into umeboshi, and cyanogenic glycosides are almost completely decomposed during the pickling process. Only trace amounts are present in the edible portion of umeboshi as well as in the mume stone. Therefore, the phenolic substances extracted using our method could be free of cyanogenic glycosides.

Plums also belong to the genus *Prunus*, and are among the most popular processed fruits. Our current findings provide a solubilization method that allows the main phenolic constituents of the pits to be extracted under mild conditions. Furthermore, this technique has potential applicability to other *Prunus* stone fruits (for example, plums, apricots, peaches, and cherries).

## 5. Conclusions

Our results demonstrate the effectiveness of subcritical methanol extraction for the solubilization of stones of *P. mume* under mild conditions. Unlike the subcritical water extract, considerable amounts of phenolic compounds such as syringaldehyde, sinapyl alcohol, and methyl coniferyl alcohol which are basic structural motifs of lignin were found in the subcritical methanol extract. Thus, it is expected that subcritical methanol extraction can be applied to easily obtain these phytochemicals from mume stones, providing a new use for this waste by-product. Further investigations of these phenolic compounds are required to determine their utility in the pharmacological and medical fields.

## Figures and Tables

**Figure 1 biomolecules-10-01047-f001:**
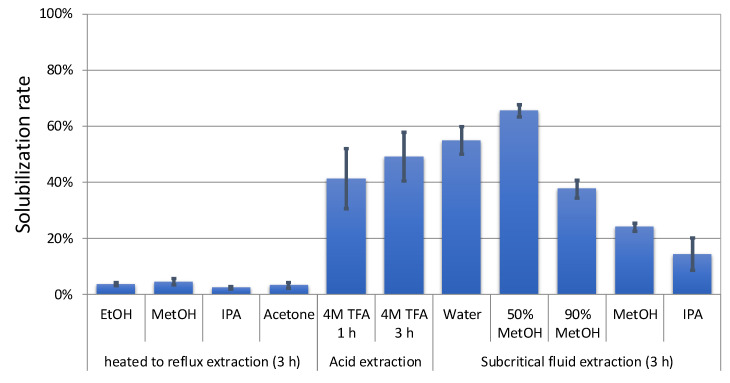
Solubilization rates of mume stones by various methods. Values are means ± standard deviations (*n* = 3).

**Figure 2 biomolecules-10-01047-f002:**
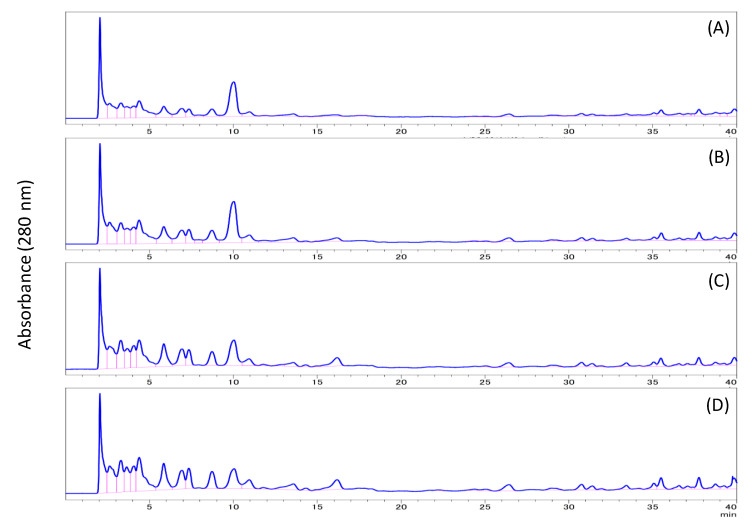
HPLC chromatograms of (**A**) methanol-soluble fraction, (**B**) ethanol-soluble fraction, (**C**) isopropyl alcohol-soluble fraction, and (**D**) acetone-soluble fraction.

**Figure 3 biomolecules-10-01047-f003:**
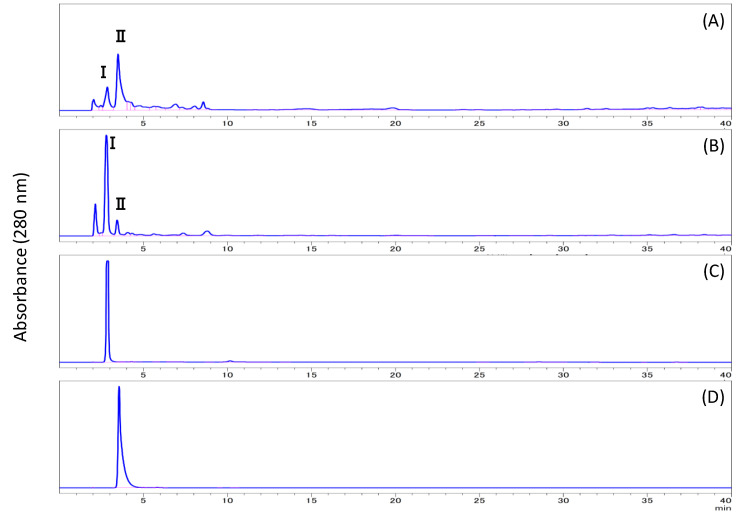
HPLC chromatograms of (**A**) trifluoroacetic acid-soluble fraction, (**B**) subcritical water extract, (**C**) 5-(hydroxymethyl)-2-furfural, and (**D**) furfural.

**Figure 4 biomolecules-10-01047-f004:**
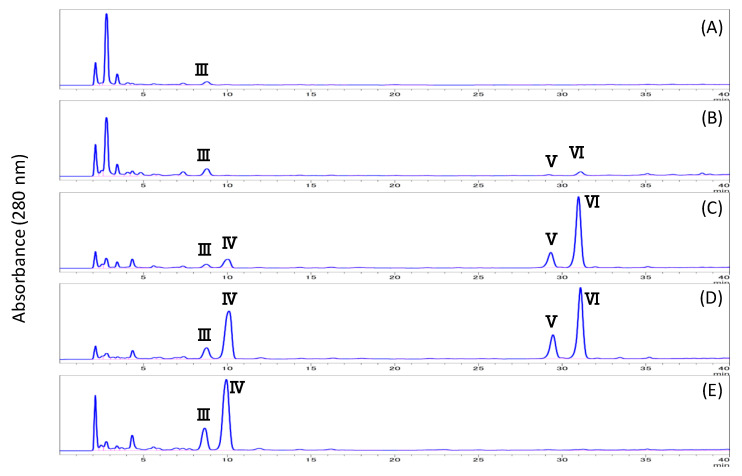
HPLC chromatograms of (**A**) subcritical water extract, (**B**) subcritical 50% methanol extract, (**C**) subcritical 90% methanol extract, (**D**) subcritical methanol extract, and (**E**) subcritical isopropyl alcohol extract.

**Figure 5 biomolecules-10-01047-f005:**
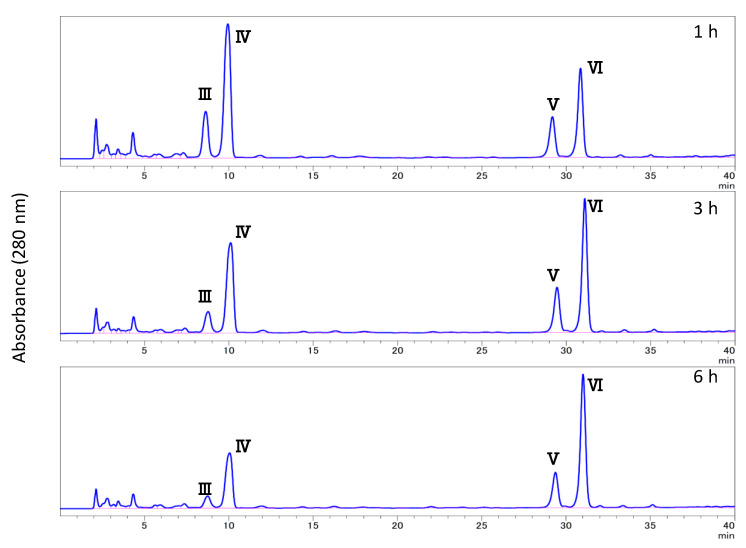
Effect of treatment time on the subcritical methanol extract.

**Figure 6 biomolecules-10-01047-f006:**
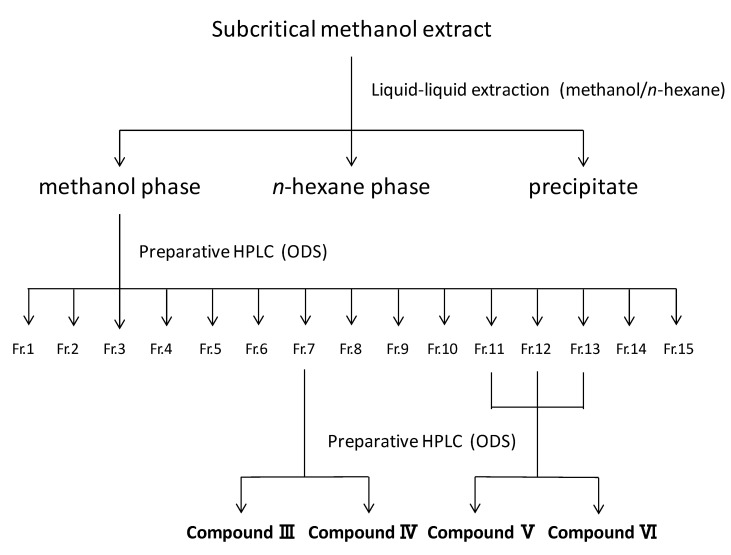
Schematic of the procedure used to isolate compounds III, IV, V, and VI.

**Figure 7 biomolecules-10-01047-f007:**
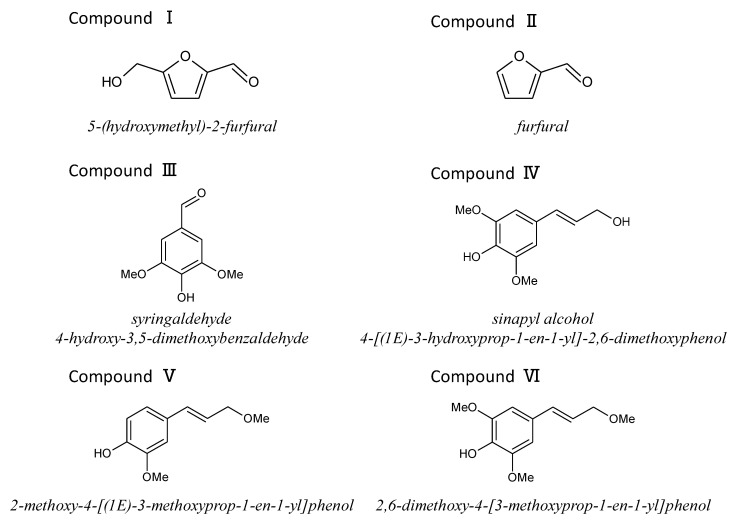
Chemical structures of compounds I–VI.

**Figure 8 biomolecules-10-01047-f008:**
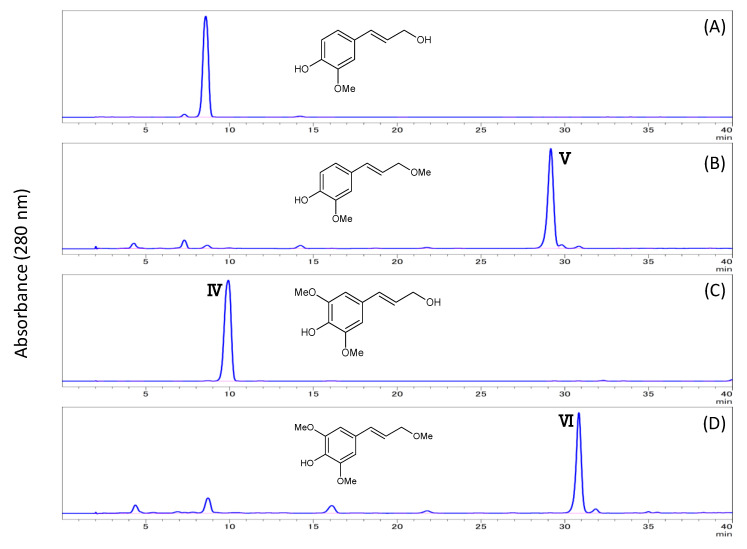
Methylation of coniferyl and sinapyl alcohols in subcritical methanol. HPLC chromatograms showing (**A**) coniferyl alcohol before subcritical methanol treatment, (**B**) the appearance of coniferyl alcohol methyl ether after subcritical methanol treatment, (**C**) sinapyl alcohol before subcritical methanol treatment, and (**D**) the appearance of sinapyl alcohol methyl ether after subcritical methanol treatment.

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
