# Peer review of "Subcritical Methanol Extraction of the Stone of Japanese Apricot Prunus mume Sieb. et Zucc."

_biomolecules, 2020, doi:10.3390/biom10071047_

Round 1
Reviewer 1 Report
Comments for the authors:
Introduction must be revised by including more recent references and new findings on the usage of subcritical fluid extraction method for the proposed study. The authors are advised to make a comparison with other similar works.
Line 202, 203: The sentence must be changed to past tense.
The disadvantages and limitations of the proposed methods must be presented in detail in the Discussion section.
Figures 1, 7, and 8 are not legible. Their quality must be improved and the font size be increased.
In Figure 4, (D) is missing. It must be corrected.
The conclusions must be improved as supported by the results obtained.
The English language and grammatical errors must be revised throughout the manuscript.
Author Response
ntroduction must be revised by including more recent references and new findings on the usage of subcritical fluid extraction method for the proposed study. The authors are advised to make a comparison with other similar works.
We added Line 55-60 and 74-77, and we also revised "Discussion".
Line 202, 203: The sentence must be changed to past tense.
We changed them to past time.
The disadvantages and limitations of the proposed methods must be presented in detail in the Discussion section.
Line 288-289, Line 301-306
Figures 1, 7, and 8 are not legible. Their quality must be improved and the font size be increased.
We improved them.
In Figure 4, (D) is missing. It must be corrected.
Thank you for your indication. We revised it.
The conclusions must be improved as supported by the results obtained.
Line 343-350
The English language and grammatical errors must be revised throughout the manuscript
We revised them as much as possible.
Reviewer 2 Report
The manuscript has been well organized. While, there are couple of questions need to be addressed before it is good to publish.
- Line 225, section 3” Results and Discussion”, while in line 286, section 4, there is another “Discussion”. In my opinion, the discussion in section 4 is kind of repeat with the introduction section, I think it is better to make the section 4 more concise or just remove the section 4 from the manuscript.
- Line 194, “2.11. MS and NMR analysis”. In this section, the author mentioned about the MS and NMR spectra. It is better to show the spectra and discuss them in the followed context, Otherwise, I don’t see the point to mention it in only section 2,”Materials and methods”.
- Line 129-131, I think it is better to present the equation 1 in a more professional format.
- In section 3, line 227-234, the tested result for lignin is 46.5%, and holocellulose is 72.5% in mume stone. So the total chemical component content (lignin + holocellulose) is 119%, which is much higher than the theoretical value (100%), Could the authors explain more about the discrepancy?
- In fig 4 (C, E, F), in between retention time 4-5min, there is a big peak which the author didn’t mention it in the paper, it is better to identify or assign it in the revised manuscript. BTW, there is a missing of " D" in Fig.4, please correct it.
- In Table 1, Hemicellulose is a part of holocellulose, so the parallel to lignin is holocellulose, not hemicellulose. The hemicellulose should be the one after Holocellulose, just like acid insoluble lignin.
Author Response
1.Line 225, section 3” Results and Discussion”, while in line 286, section 4, there is another “Discussion”. In my opinion, the discussion in section 4 is kind of repeat with the introduction section, I think it is better to make the section 4 more concise or just remove the section 4 from the manuscript.
Thank you for your indication. We revised "Results" and "Disucussion".
2. Line 194, “2.11. MS and NMR analysis”. In this section, the author mentioned about the MS and NMR spectra. It is better to show the spectra and discuss them in the followed context, Otherwise, I don’t see the point to mention it in only section 2,”Materials and methods”.
Line 249-250
3. Line 129-131, I think it is better to present the equation 1 in a more professional format.
We improved it. Line 135
4. In section 3, line 227-234, the tested result for lignin is 46.5%, and holocellulose is 72.5% in mume stone. So the total chemical component content (lignin + holocellulose) is 119%, which is much higher than the theoretical value (100%), Could the authors explain more about the discrepancy?
Your indication are just as you point out. We eliminated the analysis of cellulose and Table 1.
5.In fig 4 (C, E, F), in between retention time 4-5min, there is a big peak which the author didn’t mention it in the paper, it is better to identify or assign it in the revised manuscript. BTW, there is a missing of " D" in Fig.4, please correct it.
Line 254-258
In Table 1, Hemicellulose is a part of holocellulose, so the parallel to lignin is holocellulose, not hemicellulose. The hemicellulose should be the one after Holocellulose, just like acid insoluble lignin.
We eliminated analysis of cellulose and table 1.
Reviewer 3 Report
The aim of this study was to develop a milder and more effective method to dissolve mume stones. Therefore, author investigated the use of subcritical fluid extraction for the isolation of phenolic compounds from the stones. In order to develop a method, as the authors claim by declaring their aim, I recommend to investigate more parameters of the extraction, such as: effect of the extraction temperature, the effect of the extraction time, the effect of solid:solvent ratio. They only performed the extraction at 190oC, maybe it works even at a lower temperature. How did they choose this temperature? The extraction time could be also lowered? Is crucial, as the authors claim, to avoid degradation of phenolic by exposing them at lower temperatures for a shorter period of time during the extraction process.
Even more, the phenolic profile of the mume stones has already been investigated. The authors affirm that “Recently, mume stones were auto-hydrolyzed by microwave heating to extract polysaccharide and phenolic compounds. [3] Although considerable amounts of carbohydrate and phenolic compounds were extracted from the stones at 230 °C, extensive degradation of the phenolic compounds also occurred. Therefore, the phenolic components in the microwave-heating extract have not been identified.” (lines 313-317). The affirmation is not sustained by the results of the cited study, Seven phenolic compounds were identified in that research, some of them also identified in the present study. It is not clear which is the novelty of the present study. Neither the extraction method (subcritical fluid extraction) nor the phenolic profile of the mume stones are novel, so I cannot recommend the publication of the study in the present form. It could be published (and bring some novelty in the field) if eventually the extraction condition would be optimized and if the results would be better than those previously published.
Author Response
The aim of this study was to develop a milder and more effective method to dissolve mume stones. Therefore, author investigated the use of subcritical fluid extraction for the isolation of phenolic compounds from the stones. In order to develop a method, as the authors claim by declaring their aim, I recommend to investigate more parameters of the extraction, such as: effect of the extraction temperature, the effect of the extraction time, the effect of solid:solvent ratio. They only performed the extraction at 190oC, maybe it works even at a lower temperature. How did they choose this temperature? The extraction time could be also lowered? Is crucial, as the authors claim, to avoid degradation of phenolic by exposing them at lower temperatures for a shorter period of time during the extraction process.
Thank you very much for your indication. We added Line 301-306 in "Discussion".
Even more, the phenolic profile of the mume stones has already been investigated. The authors affirm that “Recently, mume stones were auto-hydrolyzed by microwave heating to extract polysaccharide and phenolic compounds. [3] Although considerable amounts of carbohydrate and phenolic compounds were extracted from the stones at 230 °C, extensive degradation of the phenolic compounds also occurred. Therefore, the phenolic components in the microwave-heating extract have not been identified.” (lines 313-317). The affirmation is not sustained by the results of the cited study, Seven phenolic compounds were identified in that research, some of them also identified in the present study. It is not clear which is the novelty of the present study. Neither the extraction method (subcritical fluid extraction) nor the phenolic profile of the mume stones are novel, so I cannot recommend the publication of the study in the present form. It could be published (and bring some novelty in the field) if eventually the extraction condition would be optimized and if the results would be better than those previously published.
Your indication are jut as you point out. So we revised "Discussion".
Reviewer 4 Report
All my questions, corrections, suggestions, etc., are in text of your manuscript. Read it, please.
Author Response
Thank you for your indication. We revised them as much as possible.
Round 2
Reviewer 3 Report
I don't consider that authors answered adequately to the concerns regarding the results of the present study . They added lines 301-306 to the Discussion chapter, but they still did not explained why they chose those extraction parameters. They only stated that they performed the extraction applying different extraction parameters They affirm: "On the other hand, to avoid degradation of phenol compounds, subcritical methanol extraction at lower temperature for a short period of time during extraction process was tried preliminarily. It was clarified that the solubility of stones was significantly decreased " Where do authors present the data regarding these affirmations/conclusions?
Regarding the novelty of the study, which in my opinion lacks, the authors affirm that they revised the Discussion part of the manuscript, but I don't consider that they clarified this issue.
So I still do not consider that the manuscript deserves publication.
Author Response
Thank you for your pointing it out.
Our preliminarily data were as below. The HTRS sample was treated with subcritical methanol at 130 C, 160 C or 190 C for 3 h. We did not measured each solubility by weighting method, but measured total phenolics content in each sample by the Folin-Chiocalteu method,
Temperature; absorbance at 750nm was 130 C; 0.031±0.01, 160 C; 0.110±0.02, and 190 C: 1.644±0.108.
At 190 C, we observed significant extraction from the stones. Therefore, we selected 190 C as the extraction temperature. We apologize for what we don’t have the adequate data that were pointed out by Reviewer 3.
Reviewer 4 Report
I have, again, some corrections and questions. See enclosed file, please.

Author Response
The enclosed file from Reviewer 4 was the original manuscript. However, we already sent you the revised manuscript (Ver. 1) on July 3. 2020. So, by using Ver 1, we corrected the point that were pointed out by Reviewer 4.
